# The Myers–Briggs Personality Types of Veterinary Students and Their Animal Ethical Profiles in Comparison to Criminal Justice Students in Slovenia

**DOI:** 10.3390/vetsci9080441

**Published:** 2022-08-19

**Authors:** Valentina Kubale, Branko Lobnikar, Miha Dvojmoč

**Affiliations:** 1Institute of Preclinical Sciences, Veterinary Faculty, University of Ljubljana, 1000 Ljubljana, Slovenia; 2Policing and Security Studies, Faculty of Criminal Justice and Security, University of Maribor, 1000 Ljubljana, Slovenia

**Keywords:** Myers–Briggs Type Indicator, personality, veterinary faculty students, faculty of criminal justice and security students, ethics dilemma, ethical profiles

## Abstract

**Simple Summary:**

Personality is a relatively constant and exclusive set of mental, behavioral, and physical characteristics that distinguish individuals. Personality assessment is a well-researched scientific field in which a number of theories have been developed to interpret diverse personality types. The Myers–Briggs Model (MBTI) is the most widely used self-assessment tool in the world to identify the different personality types of individuals. Since personality, with its unique characteristics, distinguishes people from each other, it also influences communication between different types of people and has an impact on attitudes toward animals. The purpose of this study was to examine the MBTI types of veterinary students compared to criminal justice students. In addition, we were interested in the attitudes of both types of student toward animals. Overall, the MBTI types and preferences of veterinary students were very different. Introversion, Sensing, Thinking, and Judging were the predominant preferences. When ethical profiles related to animals were examined, the viewpoints differed significantly between the comparison groups. The utilitarian viewpoint was most veterinary students, and the animal rights perspective was most prevalent in criminal justice students. This study highlights the importance of personality traits for better communication and work in veterinary science as well as criminal justice.

**Abstract:**

Personality types are related to trustworthy, reliable, and competent communication, especially when dealing with clients. Therefore, the purpose of this study was to investigate whether the Myers–Briggs (MBTI) indicator could be used to detect differences in the personality preferences of students at the Veterinary Faculty, University of Ljubljana (VS) compared to students at the Faculty of Criminal Justice and Security, University of Maribor (CJS). Our aim was to profile the two cohorts of students in Slovenia, to compare profiles of students from the social and natural sciences with similar personality traits, and to compare them with published results. CJS are considered well-established, well-studied, non-science ombudsman profiles of students in Slovenia for whom care and safety will play important roles in their future work, similar to VS. Views of people’s duties to animals and the implications for animal care, safety, and welfare are also very important, especially for VS. For this reason, we tested the ethical viewpoints of the two cohorts of students using an interactive web-based program. Our results show that both VS and CJS had different MBTI types, with ISTJ (Introversion, Sensing, Thinking, and Judging) preferences predominating, followed by INFJ (Introverted, Intuitive, Feeling, and Judging) in VS and ESTJ (Extraverted, Observant, Thinking, and Judging) in CJS. Between the two cohorts, the ratio between ISFJ and INFJ was statistically different. In the animal ethics study, the utilitarian viewpoint was most prevalent and statistically higher in VS compared to CJS, where the animal rights perspective was most prevalent. Compared to previous profile studies, some differences were found that could be related to the COVID-19 pandemic and/or the different generations of students. Overall, this study highlights the importance of personality traits for better communication, work, and animal research in veterinary science as well as criminal justice.

## 1. Introduction

Personality is a relatively consistent and unique set of mental, behavioral, and physical characteristics that distinguish individuals from one another [1,2]. A personality can be viewed as the sum of the characteristics that define and distinguish us from others. Individuals develop certain traits, most of which they gain or lose through interpersonal relationships with the environment and through the influence of learned patterns of behavior in their lives and behavior of others.

Personality assessment is a well-researched scientific field in which researchers have developed many theories for interpreting the various personality types. The Myers–Briggs Model (MBTI) is the most widely used self-assessment instrument in the world [3,4]. Here, personality traits are classified according to a typology of temperaments based on the work of Carl Jung [5] and his assumption that differences among people arise from their different perceptions of information about the world and themselves and from different reactions and decisions based on the information known. This includes the process of perception and the process of testing [2,6,7].

Jung [8] defined extraversion as a focus on an object with a steady flow of mental energy (libido) toward an external object, whereas the introverted type is the opposite, with energy directed inward toward the subject [8]. Jung also introduced two rational and two irrational functions in his theory. Rational functions are characterized by being active, reasonable, logical, and consistent, whereas irrational functions depend on either external objects or internal conditions [8]. Jung defined the two rational functions as thinking and feeling. He defined thinking as an activity of reasoning and feeling as an understanding of emotional logic through sense values. Jung defined the irrational functions as perception and intuition. Perception is understood as the sensory reception of stimuli, while intuition is a process of perception at an unconscious level [9]. In his theory, Jung does not advocate for a value judgment on the qualities described above but argues that both qualities are (more or less) necessary for an individual’s existence and function in all individuals.

Since Jung [8] first described his psychological types, there have been many changes to his system and much research performed. For example, the model used in the current study was designed by Isabel Myers and her mother Katharine Cook Briggs during World War II [4,9], and they hypothesized that understanding personal preferences could help with career entry. The MBTI is very similar to Jung’s original formulation, except it now has the added dimensions of judgment and perception. The MBTI thus defines four dimensions/preferences resulting in 16 possible personality types. The Myers–Briggs approach is based on the assumption that a personality consists of four basic preferences, each of which has two alternatives to choose from: Extraverted (E) or Introverted (I); Sensing (S) or Intuitive (N); Thinking (T) or Feeling (F); Judging (J) or Perceiving (P) [5]. Extraverted (E) individuals prefer to direct their energy toward other people and activities and are sociable and action-oriented, while Introverted (I) individuals prefer the inner world, the world of thoughts and ideas, and are more reserved and reflective. Sensing (S) individuals are often described as more observant and practical, while Intuitive (N) individuals tend to be more imaginative and abstract. Thinking (T) individuals are more analytical and justice-oriented, while Feeling (F) individuals are often compassionate and harmony-oriented. Judging (J) individuals tend to make decisions more quickly and are often systematic and organized, whereas Perceiving (P) individuals prefer to take more time to gather information and are often curious and adaptive [7] (Table 1).

On this basis, Myers and Briggs created a combination of sixteen different personality types, which Kiersey divided into four main groups, taking into account the characteristics of certain personalities [10]. These four groups are the Sensing-Perceiving (SP) type, the Perceiving-Judging (SJ), the Intuitive-Sensing (NF), and the Intuitive-Thinking (NT) [10].

Sensing-Perceiving (SP) individuals are adaptable, athletically and artistically oriented, relaxed, and calm and like to enjoy life. Characteristically, they are constantly looking for new opportunities and challenges, as freedom of choice is important to them. They have a sense of reality and are open-minded, tolerant, and unprejudiced. Kiersey [10] referred to this group of personalities as artisans. Perceiving-Judging (SJ) individuals are reliable, fact-oriented, accurate, conservative, hardworking, and stable. They tend to regiment their activities and want to keep their needs within certain limits. Everything must be in its place; and every action must be carried out correctly and in a controlled manner. Kiersey named this group of personalities guardians. Intuitive-Sensing (NF) individuals are very humane, believing, subjective, creative, and understanding. Character-wise, they are especially committed to bringing meaning and integrity to life, are very personally involved, and strive to always project a positive image. Kiersey referred to this group of personalities as idealists. Intuitive-Thinking (NT) individuals are abstract, analytical, curious, precise, impersonal, logical, and resourceful. They are research-oriented and extremely systematic and make decisions in a rational manner. Kiersey [10] referred to this group of personalities as rationalists (Table 2).

According to Kiersey [10], each group is divided into four (sub)groups, in conjunction with other personality dimensions, and so there are 16 social roles in the Myers–Briggs system (Table 3).

Researchers (SP), in combination with other personality dimensions, are classified into the following groups [10]: entrepreneur, promoter (ESTP), virtuoso, realist, craftsman (ISTP), entertainer, joker (ESFP), adventurer, aesthete (ISFP). Sensitive (SJ), in combination with other personality dimensions, are grouped into: director, supervisor (ESTJ), logistics, inspector (ISTJ), consul, caretaker, assistant (ESFJ), and defender, protector (ISFJ). Diplomats (NF), in combination with other personality dimensions, are classified into the following groups: protagonist, teacher, sage (ENFJ), advocate, consultant (INFJ), activist, visionary (ENFP), and mediator, dreamer (INFP). Analysts (NT) are grouped, in combination with other personality dimensions, as commander, leader (ENTJ), architect, spiritual leader (INTJ), discussant, innovator (ENTP), or logic wizard (INTP). The MBTI test has been used in a number of studies and populations, including students from the Faculty of Criminal Justice and Security (CJS), who have been studied in depth around the world as well as in Slovenia, helping to describe the personality profiles of police and security personnel. As this group has been studied in depth, it can be used as a model and also comparison in our study.

The MBTI profiling of veterinary students (VS) has been conducted in only a few studies in the literature. The published works in the area of medicine mostly involve medical students, dental medicine students, nurses, pharmacy students, or students of other medical health professions, as described by Goetz et al. [11]. In Slovenia, the MBTI profiles of VS have not been studied yet.

Interestingly, the criminal justice and veterinary medicine professions have similar tasks to perform in some respects, and both can be seen as having ombudsman profiles, where caring and safety play important roles. A comparison of the similar personality traits of social science and natural science professionals has not yet been conducted. Since the profiles of social sciences students are much better researched, it is important to combine observations to learn more about veterinary students and their MBTI profiles to advance our initial study process and improve career guidance to better address the various interests of such individuals in the field of veterinary medicine. Because personality, with its unique characteristics, distinguishes people from one another, it also influences communication between different types of people. Communication is extremely important in veterinary medicine and has recently received a growing amount of attention, in addition to the fact that a high level of knowledge and expertise are strongly promoted in this field. On the other hand, understanding the relationship between personality type and the preferred teaching methods in the classroom has also received increasing attention recently, allowing professors and assistants to use teaching methods that go beyond the traditional lecture.

Thus, the aim of our study was to examine VS in Slovenia using the MBTI tool, then compare the obtained results with the social sciences profiles of CJS with similar personality traits, and discuss the significance of the findings. We also sought to compare our results with published VS MBTI profiles in other countries and discuss possible differences in relation to the COVID-19 pandemic. In terms of similar abilities, we also examined both profiles in terms of their ethical attitudes and stances toward the treatment of animals and animal experimentation. We achieved that by exploring what ethical viewpoints regarding animals were held by both cohorts and the extent to which the two student groups were similar or different, which also allowed us to examine the MBTI preference profile in relation to animals and not just from an interpersonal skills perspective.

## 2. Materials and Methods

### 2.1. MBTI Questionnaire Study Design

The study was conducted in collaboration with a cohort of students from the Veterinary Faculty, University of Ljubljana, Slovenia (VS) and a cohort of students from the Faculty of Criminal Justice and Security, University of Maribor, Slovenia (CJS). Questionnaires were sent to both cohorts, or they were asked to fill them out at the faculties. A total of 175 VS and 138 CJS completed the questionnaires, which contained the Keirsey Temperament Classifier, a shortened version of the Myers–Briggs Temperament Indicator (MBTI). The Keirsey scale contains 70 statements with two response options, with only one response possible for each statement. Evaluation of these responses then allows assignment to one of the sixteen personality types described in the introductory section. The questionnaire was given in the Slovenian language. The VS and CJS students who completed the questionnaire were different generations of students. Participation was voluntary, and a questionnaire took 15 to 20 min to complete. After completion, the students returned the questionnaires; in the analyses, the MBTI preference scales were treated as dichotomous variables, and MBTI type was treated as a nominal variable.

### 2.2. Animal Ethics Study Design

A total of 190 VS and 124 CJS completed the questionnaire on the interactive web-based program Animal Ethics Dilemma website http://www.aedilemma.net/home (accessed on 6 July 2022) [12]. The questionnaire contained 12 questions, designed to help users gain a better understanding of their own ethical views and those of others. All statements in the program corresponded to different ethical perspectives, such as those based on contractarianism, utilitarianism, animal rights, relational perspectives, and respect for nature. It is a computer-based teaching tool designed primarily for VS. This program was structured as a computer-based role-playing game with a series of case studies that the user could either ‘play’ or explore. As a ‘provocation engine’, the tool on the website encourages the user to think critically about the viewpoint (contractarianism, utilitarianism, animal welfare, relational viewpoint, respect for nature, or hybrid) in relation to various ethically problematic situations and to defuse the user’s personal viewpoint based on the attraction of alternative viewpoints. The user experiences a role-playing situation with a set of alternative choices. The choices made by the user lead to new dilemmas in which further decisions must be made that challenge the user’s initial responses. These dilemmas help to clarify the ways in which ethical arguments relate to the situations described. The students were asked to complete a questionnaire and choose between the different answers offered according to their first thoughts.

### 2.3. Data Analysis

For the MBTI questionnaire and animal ethics study, Cohen’s d test, which uses the pooled standard deviation of compared groups, was used to verify independent samples effect sizes and to enable supplementary testing of the differences in statistical significance and comparison of the size of the effect. The major hypotheses involving comparisons between VS and CJS with regard to their MBTI profiles as well as those in the animal ethics study design were tested with Pearson’s chi-square (χ^2^) test, and post hoc comparisons were tested with a paired sample Student *t*-test. Data were analyzed using the Graph Pad Prism (8.3.0) or IBM SPSS Statistics v.28 program. The hypotheses were tested at *p*-value < 0.05, which was considered statistically significant.

## 3. Results

### 3.1. Study Design and Sample of MBTI Questionnaires

We received 175 questionnaires back from the VS and 138 from the CJS. Of the 313 returned questionnaires, only 1 questionnaire was incomplete, representing 0.3% of all returned questionnaires, or 0% in the case of CJS and 0.57% in the case of VS, which corresponded to 58.3% of students of all generations at the Veterinary Faculty and 62.7% of the students from Faculty of Criminal Justice and Security (Table 4).

In analyzing the questionnaires, we considered only those that were valid and, thus, excluded incomplete questionnaires and those in which the dimensions seemed to have the same values in the groups, e.g., the same score for extraversion and introversion, so that it was impossible to assign a person to the appropriate dimension. In the case of such a student, it would in theory be possible to repeat the completion of the questionnaire in order to determine the dimensions, but this was not possible in our case, because the questionnaires were completed anonymously. Of the 294 valid and returned questionnaires, 18 were completed with equal scores in the groups, which corresponded to 5.75% of the students (6.85% of VS and 4.34% of CJS), and these were excluded from the analysis. The data for each main group are presented below, taking into account the total number of valid questionnaires: in the case of VS, n = 162 (162 = 100%), and in the case of CJS, n = 132 (132 = 100%).

Further on, we calculated the Cohen’s d coefficient and obtained data of the size of the effect of each MBTI personal trait in the VS and CJS cohorts (Table 5). The effect size was between 2.245 and 4.069 and was largest for the Thinking (T) and Feeling (F) personal traits and lowest for the Extraverted (E) and Introverted (I) personal traits.

### 3.2. MBTI Distribution among VS

The most common preferences of VS were Introverted versus Extraverted; Sensing versus Intuition; Thinking versus Feeling; and Judging versus Perceiving, i.e., the ISTJ group, which was represented by 26.81% of the respondents. Other frequently represented types were INFJ (13.77%) and ESTJ (13.05%), while the other groups followed with lower percentages, with three groups represented by only 0.72%—ISTP, ISFP, and ESTP (Table 6). The majority of VS students were characterized by a preference for the Introverted (65.21%) and Judgment dimensions (79.71%), while the scores obtained for the Thinking/Feeling (56.52%/43.48%) and Sensing/Intuition (56.52%/43.48%) dimensions were close in percentage terms. Interestingly, the results show that 56.52% of the students gain information based on empirical facts and 43.48% based on intuition. The percentage of students with a preference to make judgments strongly prevailed (79.71%). A total of 34.79% of students belonged to this group, and 56.52% made decisions based on reasoning.

### 3.3. MBTI Distribution among CJS

The most common preferences among CJS were Extraverted versus Introverted; Sensing versus Intuitive; Thinking versus Feeling; and Judging versus Perceiving, i.e., ESTJ, which was represented by 20.74% of the responses. Surprisingly, the ISTJ group was most represented, at 27.04%, similar to the figure seen for the VS students (Table 7). The most prevalent types continued to be the ESTJ (20.74%) and ISFJ groups (11.72%). All other groups were represented with a lower percentage (between 1% and 10%). However, it is interesting to note that no student was represented in the ISFP and INTP groups. The results also showed that 73.87% of the students obtain information based on empirical facts and only 26.13% based on intuition. There was also a high percentage of students with a preference for making judgments (82.88%), with 52.25% of students belonging to this group, and 62.16% making decisions based on reasoning.

### 3.4. Comparison of VS and CJS Profiles

Table 8 shows a comparison between the CJS and VS. In both samples, there is not an even distribution between the 16 possible groups. In both samples, the ISTJ personality group predominated (27.04% in CJS and 26.81% in VS); ESTJ and ISFJ predominated among CJS, while they were also strongly represented in VS. The opposite was observed for the INFP and INTJ groups, which were represented at a higher percentage in the VS cohort, while they were in the minority among the CJS cohort. The other groups were represented at lower percentages in both samples. The ISFP and INTP groups were not represented in the sample of CJS, while all groups were represented in the sample of VS but with lower percentages.

We also compared different groups according to the basic types, which Kiersey [10] classified into four main groups, taking into account some personality characteristics (Table 9). We discovered that in both samples the most represented type was the perceiving-judging (SJ) type, which is characterized by individuals being mainly reliable, accurate, and stable. The second most represented type in both samples was the intuitive-sensing type (NF), which was more represented in the VS (28.26%) than in the CJS (16.22%). The intuitive and the perceiving types were represented in both samples and were very similar. We also analyzed the presence of the STJ personality type, which was predominant in both samples, consistent with previous research findings. There was a significant difference in two personality types between VS and CJS. We found that only the share between the ISFJ and INFJ types was statistically different (chi-square = 33,212; *p* < 0.05) between CJS and VS by performing Pearson’s chi-Square test. CJS had statistically more of the ISFJ type, and VS had more of the INFJ type. We can thus see that the difference was mainly in the S-N combination.

In order to determine the distribution of MBTI personality traits among VS and CJS, an independent samples *t*-test was performed, where statistically significant differences were observed only in the J and P traits (Table 10).

### 3.5. Animal Ethics Study Design

By defining the ethical profile in relation to animals and animal experimentation, we aimed to observe and compare the viewpoints of VS and CJS on human duties toward animals. A total of 63.33% of VS of all generations and 56.36% of CJS completed the questionnaire on the Animal Ethics Dilemma. For the animal ethics study, Cohen’s d coefficient was calculated, and the results showed the size of the effect of ethical viewpoints in the VS and CJS cohorts (Table 11). The effect size was between 8.148 and 31.370.

We found that VS and CJS held hybrid viewpoints regarding the treatment of animals and animal experimentation, but these views differed significantly. Among VS, the utilitarian viewpoint was the most prevalent (50.5%), followed by the animal welfare (26%) and the respect for nature views (18.4%). Lower percentages were found for the contractarian (1.1%) and the relational viewpoints (7.8%). The greatest differences between responses were observed for the utilitarian and animal rights viewpoints. Among CJS, the animal rights view was the most prevalent (42.5%), followed by the utilitarian (30%) and the respect for nature viewpoints (17.4%). The lowest percentages were observed for the contractarian (2.7%) and the relational viewpoints (7.4%). The greatest fluctuation between responses was also observed for the utilitarian and animal rights viewpoints (Figure 1). The utilitarian viewpoint was statistically higher in VS compared to CJS, and the animal rights perspective was significantly higher in SJS in comparison to VS, as shown in the results of Student’s and Pearson’s chi-square test (*p* < 0.05).

## 4. Discussion

Personality assessment is a well-researched scientific field in which a number of theories have been developed to interpret different personality types. The MBTI is the most widely used self-assessment instrument in the world to help identify the different personality types of individuals. Just as personality, with its unique characteristics, distinguishes people from one another, it also influences communication between different types of people. Communication is essential in veterinary medicine and has recently become even more important, and a high level of knowledge and expertise in the field is strongly encouraged in this profession.

The purpose of this study was to investigate the MBTI types of a cohort of VS in comparison with a cohort of CJS, to evaluate the previously characterized profile of CJ and to examine similarities and differences between the two profiles. The social science profile of CJS is a well-studied one, also known as the ombudsman profile. Moreover, it is the most studied profile in Slovenia [6]. Some studies suggested that the profile may change over the years, and so it was necessary to address this issue first. Twenty years after the original studies were conducted [6], the same instrument that was developed and compared with other studies was tested with CJS, as well as other students, and used with the same protocol to study the profile of VS. In this way, a proven and tested model was used to obtain information about the profile of VS. In addition to the MBTI profile, the characteristics of the ethical viewpoints toward animals were examined in both populations.

Pagon and Lobnikar [6] involved students from the Faculty of Criminal Justice and Security (CJS) and the Faculty of Social Sciences (FSS) in their study. They found that most CJS were of the SJ personality type, with 41.7% in the ESTJ group and 14.4% in the ISTJ. All other personality groups were represented by much lower percentages. Among FSS students, the ESTJ personality type was the most represented, although with a much lower percentage than observed for the CJS. In addition to the ESTJ personality type, the ENFP type was also represented at 14% and the ENTP personality type at 11%.

Henewicz [5] included 1,282 police recruits and police officers from Florida, USA in his study. The results indicated that the majority of American police officers were ‘Guardians/Sensitive’, and, among these, the most represented personalities were ESTJ and ISTJ. As many as 51.1% of the respondents belonged to the SJ group, followed by the SP group, with 26.2%, and the least represented groups were NT (31.1%) and NF (9.6%) [5]. The author concluded that the STJ group fit the profile of a ‘police personality’. The author then compared his findings with those of three different professional groups—teachers, social work students, and dental medicine students. He found that the ENFP and INFP personality types, which belong to the NF personality group, predominated in the teacher group. As many as 68.2% of the respondents belonged to this group. Similar results were also found among social science students, where the NF personality type was represented by 40.6%. Among both teachers and social work students, the STJ group (which was strongly represented in the police survey) was the least representative, at only 9% in both groups.

Among dental medicine students, the SJ group was the most represented, at 44.5%, followed by the NF personality group. The STJ group, which is most common among police officers, was represented by 27.9% of the dental medicine students. A similar study among police officers was conducted in 1984 by Lynch and McMahon, who used the MBTI instrument on a sample of 772 American police officers. The results were consistent with those of Henewicz, as their study also showed that most officers were the STJ personality type [13]. They also found that only a small proportion of the police officers studied were introverts.

According to the results of Pagon and Lobnikar [6], the results for the CJS and Henewicz’s results were very similar among police officers. In both cases, the SJ type was strongly expressed, i.e., individuals were reliable, fact-oriented, accurate, routine, stable, diligent, and persistent. Thus, the data from previous research confirmed the theory of the STJ profile as a ‘police personality’ [14,15]. Our results confirmed previous studies and predictions to some extent. Most of the CJS were of the STJ personality type, for which previous research data confirmed the ‘police personality profile’ theory. Like Lobnikar and Pagon’s research [6], we found that the profile was similar among CJS. They are characterized by exceptional precision, conscientiousness, and deliberate and careful thinking. They hate irregularity or deviation and are fiercely loyal to their organizations. Although reserved, they are affectionate, friendly, and tolerant in their interactions and communication. ESTJ was the personality type most prominent in Lobnikar and Pagon’s study [6] and ranked second in ours. It is characterized by individuals who are more expressive and concrete by nature. Students who belong to this personality type are very strict and tend to enforce rules and procedures, which to some extent is also true for the predominant personality type in our study. The results obtained in our study varied with respect to the type of student, as more introverted students were observed than extraverted ones. This observation could be related to more advanced technology and the inclusion of Generation Z students, who have grown up with access to the Internet and portable digital technology from a young age. As such, while they are known to be more educated, they are also more introverted. The observed difference could also be related to the recent COVID-19 pandemic or even a consequence of Generation Z becoming introverted due to related lockdowns. Other findings related to CJS are consistent with our predictions, as the police and police work characteristics coincide with the characteristics of the STJ personality type or so-called guardians or ombudsmen. Both the ISTJ and ESTJ types have the role of ombudsmen, who are willing to cooperate with superiors and carry out orders consistently and without error. The values they cultivate are primarily concern for life, family, work, and community. They trust the authorities or those higher in the hierarchy while committing to social responsibility in all aspects of their lives. For ombudsmen, security is crucial in all respects; they value hard work and want to maintain the existing order, and their main interests are education and constant progress, along with morality and maintaining the existing order, which strongly coincides with the order of the police organization.

The results for the MBTI types of the VS presented in our study were surprisingly similar to those of the CJS, although the latter tended to be more social science-oriented. Overall, the MBTI types of the VS in our study varied, with Introversion, Sensing, Thinking, and Judging (ISTJ) being the predominant preferences and, therefore, having similar preferences to those described above for the CJS. This finding can be supported by the characteristics of this personality type that are simultaneously consistent with the work of the criminal justice system itself and the work of veterinarians. Veterinarians have high levels of compassion, are caring, and do their jobs because they are aware of their social responsibilities. These characteristics coincide with those held by people who work in a police organization.

Comparing the VS in this study with previous published studies of the profiles of VS in the United States conducted before the COVID-19 pandemic, we can see similar differences to those seen in CJS. In a previously published study [12], a predominant ESTJ profile was found. The study found that VS had different MBTI types and preferences, with Extraversion, Sensing, Thinking, and Judging (ESTJ) preferences being the most prevalent. VS were assessed using their MBTI preferences in combination with admission mini-interview scores to assess the impacts on diversity of the applicants admitted to one of the top veterinary schools in the country—the University of California Davis School of Veterinary Medicine (UCDSVM) [15]. Important skills for future veterinarians include decision-making skills, communication skills, and attitudes toward ethical and social scenarios of the profession. In a study by Johnson et al. [16], the ESTJ-ISTJ profile was presented for such individuals. In this study, the major personality types among veterinary students were ESTJ (11.5%), ESFJ (11.1%), and ISTJ (10.3%), whereas, at the same time, the national personality types were ISFJ (13.8%), ESFJ (12.3%), ISTJ (11.6%), and ISFP (8.8%). However, in earlier years (from 1996 to 2003), other patterns were observed (ESTF and ISTF) which dominated at veterinary schools. Our results are, to some extent, similar to those of Johnson et al. [16]. In the veterinary context, ISTJ is considered to be a task-oriented personality, held by people who carefully weigh information before making decisions and who are decisive, reliable, conscientious, and maintain a no-nonsense and logical communication style. They give and expect others to follow precise instructions. On the other hand, ESTJ profiles are known for liking to solve immediate problems, using past experiences as reference points for decisions, and holding themselves and others to clear standards. In terms of communication, ESTJ profiles are quick to challenge ideas and facts and are often eager to discuss and take matters into their own hands, which can be seen as abrupt or impersonal. The more pronounced introversion profiles of our recently studied VS could also be due to a different generation of students (Gen Z) recently affected by the COVID-19 pandemic or due to the pandemic itself.

In Johnson et al. [16], the differences were associated to some extent with gender differences. Thinking preference predominated in males at veterinary school, and feeling preference predominated in females. The results suggest that personalities shift and attract different types of veterinarians. Johnson et al. [14] reported an increase in feeling types over thinking types, which is not unique to women. In our study, we identified the INFP profile as the second most common. ENFP and INFP profiles have been shown to be more associated with the female population. Although feeling type personalities are more common among females, it is possible that new male VS also tend more toward the feeling type. We wanted to examine gender differences in our study, particularly in relation to VS. However, the local Veterinary Faculty is currently strongly gender-biased, with more female students studying veterinary medicine. In fact, the ratio of male to female students is 10% and 90%, respectively, and the number of male students continues to decline. This is due to the current system where grades from high school and final high school exams are considered when entering studies, with female students performing better in the required subjects. As the University of Ljubljana, to which the Veterinary Faculty belongs, promotes gender equality and is interested in developing tools to combat gender bias in recruitment, this study is a small contribution to improving gender equality at our institution. Like some of the studies mentioned above [15], we are also considering mini-interviews that could be linked to a profile test that includes a MBTI test for entrance to our faculty.

In addition to being interested in the profile of VS and making a small contribution to gender equality, we are also interested in improving the learning process with students and their communication skills. It is known that COVID-19 had an impact on students’ learning styles, particularly affecting extraverted and introverted personality types, who also accepted virtual learning differently during COVID-19 pandemic [17]. Extraverted students were less comfortable with virtual learning than introverted students, for whom it was a useful and enjoyable experience. As the type of student has changed over the decades, knowledge of student profiles, in addition to the COVID-19 pandemic, has implications for the learning process, how it is delivered, and the amount and type of information that students absorb. The information gained about the student’s personality type provides an opportunity to incorporate this knowledge into the didactic process to develop different processes and also to design part of the process as virtual or e-learning processes or platforms.

The Myers–Briggs model also helps us understand the way we take in information, how we see things, our interests, and ideas. Since the sensitive and feeling personality types predominate among the general population, a shift toward this personality type in veterinarians could help to reduce communication problems between doctors and pet owners. In recent years, veterinary schools have increased their focus on business and communication skills because of a reported lack of veterinarians’ ability to communicate with their clients. Therefore, it is important to know the current profiles of VS to understand how they perceive and respond to the world and how to teach them relevant non-technical skills, such as communication. Also of great importance is considering how to maintain diversity in the VS population so that diverse types of veterinarians can work in farm animal practice, companion animal practice, public health, research, or academic specialties. From this perspective, it might also be useful to consider changing the required admission criteria. In the second part of our study, we explored the question of what ethical viewpoints (contractarian, utilitarian, animal rights, relational view, respect for nature, and hybrid point of view) are held in relation to animals and to what extent the two profiles of VS and CJS are similar or different, which also allowed us to examine the MBTI preference profile in relation to animals and not only from the point of view of interpersonal skills.

Briefly, the contractarian viewpoint is based on the assumption that ethical obligations originate in mutual agreements or contracts between people and that mutual cooperation is essential in reaching such agreements. When we pursue our own interests, we can benefit from the help of others, who will receive something in return. Mutual agreements and cooperation are the key. Non-human animals cannot make agreements; therefore, animals have neither consciences nor moral obligations. They lack the understanding and control needed to enter a contractual agreement. However, we may have indirect ethical obligations to animals, because they may be important to others. As a result, animals neither create nor have moral duties. From a utilitarian perspective, moral balancing is important. In deciding whether an action is morally justified, it is important to weigh the sum of all of the benefits that the action brings and compare them to the sum of all of the harmful consequences, including from the point of view of animals. According to the utilitarian perspective, the only significant ethical concerns regarding animals are animal and human welfare. Activities which have an adverse impact on the well-being of animals may be justified if, all things considered, they increase the welfare of humans or other animals [12]. According to the theory of utilitarianism [18], the following conditions must be met: (a) benefit principle (e.g., when using animals in scientific research, the general benefit must be maximized, and the animals that we use for research purposes must be provided with the most humane living conditions possible); (b) value standard (evaluation of the negative and positive aspects of, e.g., animal experiments depending on how they turn out); (c) consequence (a positive outcome for human health may outweigh the negative impact on the experimental animals); and (d) justice (all living beings involved in the experiment must be treated fairly). This is the most common viewpoint among VS.

The relational view is a group of interrelated views whose focus is on the ethical significance of relationships between animals and humans and between and among humans themselves. According to one viewpoint, our duties to animals depend on whether or not they are close to us. The animal rights viewpoint is a broad one that focuses on treating animals with respect, not harming them, or even treating them with the same rights as humans. This ethical standpoint was most highly ranked by CJS. There is a contrast here with utilitarianism, because the utilitarian viewpoint believes that in maximizing welfare or happiness, it may be morally justifiable to violate what rights advocates would call ‘rights’. Respect for nature advocates believe that we have a duty to protect not only individual animals but the species to which they belong and especially the integrity of each species. Interestingly, while CJS seem to protect animals, VS seem to care about the welfare of both people and animals. The hybrid point of view is composed of more than one viewpoint or perspective and combines different views, of which one is prominent.

Interestingly, when we examined the ethical profiles related to animals and animal experimentation, we discovered that the viewpoints of VS and CJS differed significantly, despite similar and partially overlapping MBTI profiles. Ethical viewpoints regarding animals and animal experimentation were hybrid in both studied cohorts, and, interestingly, we observed that the utilitarian viewpoint was the most represented, followed by the animal rights viewpoint. The utilitarian perspective was the most represented in VS and significantly higher than in CJS, and, on the other hand, the animal rights viewpoint was significantly higher in VS compared to CJS. Each personal profile provides us and students with information, understanding, and awareness of their ethical choices and relates them to ethical theories. Knowing a student population’s viewpoints can be helpful in order to prepare them for dialogue regarding different ethical issues, especially issues regarding animals, which are typically polarized. To prevent a lack of dialogue, it is very important that students do not lack understanding of ethical and debating skills. Such information also serves as a tool in training in bioethics. In conclusion, we would like to point out that the MBTI is a well-tested scientific method that gives us interesting and highly relevant results, but, it only measures preferences and not competencies. It should not be used as a tool for personnel selection, but it should be understood primarily as a human resource development tool [6,19]. The MBTI is a good method that any organization can use to determine the personal characteristics of its employees when they enter the organization or when they are hired. The same is true for the selection of appropriate faculty and subsequent career orientation. Some veterinary schools have also used it to select students entering the degree process [15,20].

This study emphasizes the diversity of VS and highlights their character traits, which may have been influenced in recent years by the COVID-19 pandemic as well as by technology-related differences among different generations of students. The study also highlights the importance of personality traits for better communication and collaboration in veterinary medicine and the social sciences. It also offers educators the opportunity to improve their teaching methods and career counseling resources to better reach students of all MBTI preferences and advance student-tailored teaching that adapts to students’ needs in order to increase student engagement and motivation and to achieve better performance.

## Figures and Tables

**Figure 1 vetsci-09-00441-f001:**
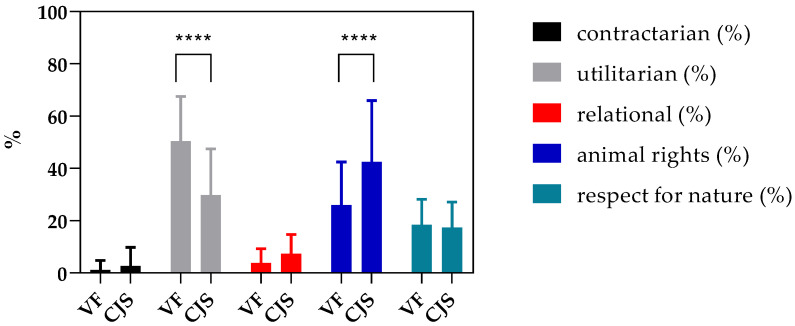
Animal ethical profile of students from the Veterinary Faculty (VS) and Faculty of Criminal Justice and Security (CJS). The ethical profile of the VS was compared with the profile of the CJS. The different viewpoints are marked in different colors: contractarian (*black*), utilitarian (*gray*), relational (*red*), animal rights (*blue*), and respect for nature (*green*). The chart shows the percentage of students holding the different viewpoints. Statistical testing was performed using an unpaired Student’s *t*-test using GraphPad Prism (8.3.0). **** *p*-value < 0.0001.

**Table 1 vetsci-09-00441-t001:** Distribution of alternative preferences in the general population (adopted from Pagon and Lobnikar [6].

Alternative Preferences	Distribution in General Population
Extraverted (E) or Introverted (I)	75–25%
Sensing (S) or Intuitive (N)	75–25%
Thinking (T) or Feeling (F)	50–25%
Judging (J) or Perceiving (P)	50–50%

**Table 2 vetsci-09-00441-t002:** Groups according to MBTI (adopted from Kiersley [10]).

Kiersey Temperament Sorter (KTS)	Modern Denomination (Summarized by 16 Personalities)
Artisan	Explorer
Guardian	Sentinel
Idealist	Diplomat
Rationalist	Analyst

**Table 3 vetsci-09-00441-t003:** Myers–Briggs temperament indicator (MBTI) (adopted from Kiersey [10]).

MBTI					
		Sensing Types (S)		Intuitive Types (N)	
		Thinking (T)	Feeling (F)	Feeling (F)	Thinking (T)
Introverted (I)	Judging (J)	ISTJ	ISFJ	INFJ	INTJ
	Perceiving (P)	ISTP	ISFP	INFP	INTP
Extraverted (E)	Perceiving (P)	ESTP	ESFP	ENFP	ENTP
	Judging (J)	ESTJ	ESFJ	ENFJ	ENTJ

**Table 4 vetsci-09-00441-t004:** Returned questionnaires from the Veterinary Faculty (VS) and Faculty of Criminal Justice and Security (CJS).

	VS	VS (%)	CJS	CJS (%)	Together	Together (%)
Total returned questionnaires	175	100%	138	100%	313	100%
Total excluded/invalid questionnaires	13	7.42%	6	4.34%	19	6.07%
Of which incomplete and thus invalid	1	0.57%	0	0	1	0.32%
Of which with mean values and thus unusable for analysis	12 (of 174)	6.85%	6 (of 138)	4.34%	18 (of 313)	5.75%
**Total returned valid questionnaires (n)**	**162**	**92.58%**	**132**	**95.66%**	**294**	**93.93%**

**Table 5 vetsci-09-00441-t005:** Independent samples effect sizes performed with MBTI profile variables by calculation of Cohen’s d coefficient.

Personal Trait	Cohen’s d	Point Estimate	95% CI Lower	95% CI Upper
**E**	2.245	−0.741	−1.117	−0.361
**I**	2.245	0.741	0.361	1.117
**S**	3.528	−0.541	−0.912	−0.168
**N**	3.528	0.541	0.168	0.912
**T**	4.069	−0.417	−0.786	−0.047
**F**	4.069	0.417	0.047	0.786
**J**	3.576	−0.340	−0.708	0.029
**P**	3.576	0.340	−0.029	0.708

**Table 6 vetsci-09-00441-t006:** MBTI preferences of the students from the Veterinary Faculty (VS).

		SENSING (S) 78 (56.52%)		INTUITIVE (N) 60 (43.48%)	
		THINKING (T) 78 56.52%	FEELING (F) 60 43.48%	FEELING (F) 60 43.48%	THINKING (T) 78 56.52%
INTROVERTED (**I**) 90 65.21%	JUDGING (**J**) 110 79.71%	**ISTJ** 37 26.81%	**ISFJ** 6 4.35%	**INFJ** 19 13.77%	**INTJ** 13 9.42%
	PERCEIVING (**P**) 28 20.29%	**ISTP** 1 0.72%	**ISFP** 1 0.72%	**INFP** 10 7.26%	**INTP** 3 2.17%
EXTRAVERTED (**E**) 48 34.79%	PERCEIVING (**P**) 28 20.29%	**ESTP** 1 0.72%	**ESFP** 4 2.90%	**ENFP** 6 4.35%	**ENTP** 2 1.44%
	JUDGING (**J**) 110 79.71%	**ESTJ** 18 13.05%	**ESFJ** 10 7.25%	**ENFJ** 4 2.90%	**ENTJ** 3 2.17%

**Table 7 vetsci-09-00441-t007:** MBTI preferences of the students from the Faculty of Criminal Justice and Security (CJS).

		SENSING (S) 82 (73.87%)		INTUITIVE (N) 29 (26.13%)	
		THINKING (T) 69 62.16%	FEELING (F) 42 37.84%	FEELING (F) 42 37.84%	THINKING (T) 69 62.16%
INTROVERTED (**I**) 53 47.75%	JUDGING (**J**) 92 82.88%	**ISTJ** 30 27.04%	**ISFJ** 13 11.72%	**INFJ** 2 1.80%	**INTJ** 4 3.60%
	PERCEIVING (**P**) 19 17.12%	**ISTP** 1 0.90%	**ISFP** 0 0.00%	**INFP** 3 2.70%	**INTP** 0 0.00%
EXTRAVERTED (**E**) 58 52.25%	PERCEIVING (**P**) 19 17.12%	**ESTP** 4 3.60%	**ESFP** 1 0.90%	**ENFP** 9 8.10%	**ENTP** 1 0.90%
	JUDGING (**J**) 92 82.88%	**ESTJ** 23 20.74%	**ESFJ** 10 9.00%	**ENFJ** 4 3.60%	**ENTJ** 6 5.40%

**Table 8 vetsci-09-00441-t008:** Comparison between Veterinary Faculty (VS) and Faculty of Criminal Justice and Security (CJS) students.

TYPE	VS	CJS
**ISTJ**	26.81%	27.04%
**ISFJ**	4.356%	11.72%
**INFJ**	13.77%	1.80%
**INTJ**	9.42%	3.60%
**ISTP**	0.72%	0.90%
**ISFP**	0.72%	0.00%
**INFP**	7.26%	2.70%
**INTP**	2.17%	0.00%
**ESTP**	0.72%	3.60%
**ESFP**	2.90%	0.90%
**ENFP**	4,35%	8.10%
**ENTP**	1.44%	0.90%
**ESTJ**	13.05%	20.74%
**ESFJ**	7.25%	9.00%
**ENFJ**	2.90%	3.60%
**ENTJ**	2.17%	5.40%

**Table 9 vetsci-09-00441-t009:** Comparison of the different groups according to the basic personality types.

TYPE	VS	CJS
**SP**	5.04%	5.41%
**SJ**	**51.45%**	**68.47%**
**NF**	28.26%	16.22%
**NT**	15.22%	9.91%
**STJ**	**39.85%**	**47.75%**

**Table 10 vetsci-09-00441-t010:** Independent samples *t*-test was used to determine the distribution of different MBTI profile variables.

Independent Samples *t*-Test					
		Levene’s Test for Equality of Variances		*t*-Test for Equality of Means	
		F	Sig.	*t*	df
**E**	Equal variances assumed	0.124	0.725	−3.971	113
**I**	Equal variances assumed	0.124	0.725	3.971	113
**S**	Equal variances assumed	2.172	0.143	−2.900	113
**N**	Equal variances assumed	2.172	0.143	2.900	113
**T**	Equal variances assumed	1.298	0.257	−2.237	113
**F**	Equal variances assumed	1.298	0.257	2.237	113
**J**	Equal variances assumed	4.747	0.031	−1.824	113
**P**	Equal variances assumed	4.747	0.031	1.824	113

**Table 11 vetsci-09-00441-t011:** Paired samples effect sizes performed with ethical viewpoints by calculation of Cohen’s d coefficient.

Ethical Viewpoint (Pair CJS—VF)	Cohen’s d	Point Estimate	95% CI Lower	95% CI Upper
Contractarian	8.148	0.170	−0.007	0.347
Utilitarian	24.939	0.854	−1.059	−0.647
Relational	8.829	0.461	0.275	0.646
Animal rights	31.274	0.542	0.353	0.730
Respect for nature	14.694	0.080	−0.256	0.097

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
