# Peer review of "The Myers–Briggs Personality Types of Veterinary Students and Their Animal Ethical Profiles in Comparison to Criminal Justice Students in Slovenia"

_vetsci, 2022, doi:10.3390/vetsci9080441_

Round 1

Reviewer 1 Report

The authors did an admirable job writing up this manuscript. Overall, it was very well-written, with some minor grammatical errors that the journal editing staff  can address. Still, I do have some concerns that the authors need to address: some major and some minor.

1.  In other related studies, the sample human cohorts were much larger. Given the nature of the study and the simplicity of the data acquisition, why did the authors not collected more data? I would like to see a power calculation to validate these sample sizes.

2. Given the variability in the trait characteristics, the Student's t- test is a lower statistical level tool. Bring in a statistician to provide a more broader and comprehensive analysis of your data. There are a number of more rigorous STAT tools available. My belief is that your results may vary when they are applied to your data.  This is a major requirement for this manuscript.

3. The abstract: Add a short statement about how the vet students compared to the criminal justice students. I thought your abstract summary statement was too vague. As written, I would not be motivated to read your article..

4. This manuscript has many abbreviations. Your  designation of your student cohorts is a bit confusing. Please simplify to something like veterinary students (VS) and criminal justice students (CJS). This inclusion of Faculty (F) is not necessary and adds an element of confusion. For example, in Figure 1 , what does the abbreviation "FFV" mean? Keeping things simple makes it easier on the reader to follow. BTW, were the error bars SD or SEM?

5. Although informative, I thought that your introduction was a bit long and could be shortened. 

6. I felt that some of the categories in the Animal ethics study design (i.e. contractarianism, utilitarianism, hybrid viewpoint) could have been better defined.

Author Response

Response to Reviewer 1 Comments

Comments and Suggestions for Authors

The authors did an admirable job writing up this manuscript. Overall, it was very well-written, with some minor grammatical errors that the journal editing staff  can address. Still, I do have some concerns that the authors need to address: some major and some minor.

Response: We thank the reviewer for such a positive opinion about our manuscript and also for suggestions, which significantly improved the manuscript.

Point 1: In other related studies, the sample human cohorts were much larger. Given the nature of the study and the simplicity of the data acquisition, why did the authors not collected more data? I would like to see a power calculation to validate these sample sizes.

Response 1:

We thank for the comment on the sample size, which prompted us to revise the manuscript substantially. We have initially submitted manuscript with lower sample size and have in last five month increased samples size and rewrite manuscript. We are dealing with a small sample size of student cohorts, who are studying at both Faculties. In the Veterinary faculty we have around 300 students in all years and at the Faculty for Criminal Justice and Security only 220. This are the numbers of students corresponding to whole numbers of students/year and are consisting of 5 generations of students in the case of Faculty of Criminal Justice and Security (CJS) and 6 generations of students in the case of Veterinary faculty (VS). Therefore our samples of participating students was corresponding to 58,3% and  63,33% for MBTI profiles study and ethics dilemma study, respectively in the case of students of Veterinary faculty (VS) and 62,7%  56,36% for the students of the Faculty for Criminal Justice and Security (CJS).

Also not all students are willing to complete the various questionnaires they receive. Nevertheless, we increased our sample size by 191% for VS and by 345% for students from student cohort of CJS for the MBTI profiling in our resubmission of the manuscript. The number of students participating was 175 and 138, respectively, representing 62.5% VS and 57.3% of CJS. For the animal ethics studies, the sample size was also increased by 691% and 490%, corresponding to 190 and 124 VS and CJS, respectively. With the significant increase in sample size, there were only minor changes in the profile tests and none in animal ethics design, where only some percentages changed and the ratios remained the same. We have also written more about the results and conclusions and hope that we have improved and met the requirements for research design, clear presentation of results, and conclusions supported by the results as suggested.

Additionally Cohen’s d test was performed for MBTI profile study and animal ethics part of the study to test sample size and more information was added to materials and methods section, as well as results section, where new data tables are inserted.

In addition odds ratios (OD) were calculated for MBTI profile study and animal ethic study. When comparing both cohorts have observed OD 1.0753 (0.8103 – 1.4270) and P = 0.6149 for the MBTI study samples and OD 0.8990 (0.6690 – 1.183), with P = 0.4234 regarding animal ethic study study and OD 1.0753 (0.8103 – 1.4270).

Point 2: Given the variability in the trait characteristics, the Student's t- test is a lower statistical level tool. Bring in a statistician to provide a more broader and comprehensive analysis of your data. There are a number of more rigorous STAT tools available. My belief is that your results may vary when they are applied to your data.  This is a major requirement for this manuscript.

Response 2: We thank reviewer for the comment, which is very relevant. We are aware regarding the importance of the statistical methods. As written, results were supplemented with Cohen’s d test, two tailed unpaired Student t-test for each personal trait and Chi-square (c2) statistics was additionally performed for MBTI part of the study and well as for the animal ethics study.

Point 3: The abstract: Add a short statement about how the vet students compared to the criminal justice students. I thought your abstract summary statement was too vague. As written, I would not be motivated to read your article..

Response 3: We thank reviewer for the suggestion. Sentence was added and abstract was rewritten substantially.

Point 4: This manuscript has many abbreviations. Your  designation of your student cohorts is a bit confusing. Please simplify to something like veterinary students (VS) and criminal justice students (CJS). This inclusion of Faculty (F) is not necessary and adds an element of confusion. For example, in Figure 1 , what does the abbreviation "FFV" mean? Keeping things simple makes it easier on the reader to follow. BTW, were the error bars SD or SEM?

Response 4: We thank reviewer for the suggestion. We have used abbreviations our Faculty normally use in everyday life. We have changed abbreviations to VS and CJS. For the FVV we apologize, thank you for spotting it. It is an abbreviation for Slovenian name of Faculty of Criminal Justice and Security, which we have changed accordingly. Error bars are SD.

Point 5: Although informative, I thought that your introduction was a bit long and could be shortened. 

Response 5: We thank reviewer for observation and suggestion. In the field of veterinary medicine (more than in the field of social sciences) this knowledge is less known and therefore the introduction is longer with more details explained. We would like this information to be available to other readers who might find all the information useful, including veterinary students who are learning this topic as part of the “Veterinary Medicine Communication 1” curriculum subject. However, we have shortened the introduction and removed some paragraphs and sentences and rephrased some parts.

Point 6: I felt that some of the categories in the Animal ethics study design (i.e. contractarianism, utilitarianism, hybrid viewpoint) could have been better defined.

Response 6: We thank reviewer for the comment. We have added more information in the text regarding categories, especially hybrid viewpoint which was not described.

Point 7: Overall, it was very well-written, with some minor grammatical errors that the journal editing staff  can address.

Response 8: We have used professional English editing service to improve the manuscript English language.

Reviewer 2 Report

In this manuscript, Kubale et al. investigated the MBTI types of cohort of veterinary students in comparison to cohort of criminal justice students to evaluate the previously characterized profile of criminal justice student and to examine similarities and differences between the two profiles. The MBTI is a well-tested scientific method giving people interesting and highly relevent results. This study emphasizes the diversity of veterinary students and highlights their character traits. The article was written well and ready to be published. 

Author Response

Response to Reviewer 2 Comments

Comments and Suggestions for Authors

In this manuscript, Kubale et al. investigated the MBTI types of cohort of veterinary students in comparison to cohort of criminal justice students to evaluate the previously characterized profile of criminal justice student and to examine similarities and differences between the two profiles. The MBTI is a well-tested scientific method giving people interesting and highly relevent results. This study emphasizes the diversity of veterinary students and highlights their character traits. The article was written well and ready to be published. 

Response: We appreciate the reviewer´s enthusiasm for our study and we thank the reviewer for such a positive opinion about our manuscript.

This manuscript is a resubmission of an earlier submission. The following is a list of the peer review reports and author responses from that submission.

Round 1

Reviewer 1 Report

Comparing the personality profiles between VF and FFCJS students  and relating it to animal ethics issue is an acceptable interest point. Overall, the manuscript read well, with some minor errors noted.

The major weakness of this manuscript is the small sample size within and between these 2 study cohorts. I would recommend that the authors delay their manuscript submission and gather more data from multiple classes (i.e. 5 classes). You don't have the sample pool size needed to support your conclusions. I would have been more accepting had I reviewed data consisting >200 response per study cohort.

Reviewer 2 Report

General Comments:

Thank you for asking me to review the paper entitled, Myers-Briggs Personality Type Indicator of Veterinary Students in Alliance with their Ethical Profiles Regarding Animals in Comparison to Criminal Justice Students in Slovenia.  I feel the paper provides some additional value to the current literature.  It does require extensive grammatical editing.  I have some specific suggestions which I feel could improve the paper.

Specific Comments:

  1. Title: Veterinary is misspelled.
  2. Introduction:
    1. Consider merging Table 2 and Table 3 if you think this is feasible. I think it would help connect the Modern denomination with MBTI.
    2. The introduction might be able to be streamlined some. First paragraph can probably be reduced in wording, and some of the second paragraph seems to be redundant with Table 1.
  3. Materials & methods:
    1. Were other questionnaire’s considered? Could the authors explain why the Keirsey Temperament Classifier was used? 
    2. I was a little confused about the animal ethics study design. I didn’t see any mention of it in the Introduction, so it was confusing to see it come up in the methods.
  4. Discussion:
    1. Please list the limitations of the study (e.g., small sample size, generalizability etc.)
    2. The authors allude to gender differences in Johnson et al (reference 14), were any demographics recorded? Thoughts on how gender may affect results?